# Effects of Mood on Psychophysiological Detection of Concealed Information and the Relation to Self-Assessed Lying Ability

**DOI:** 10.3390/brainsci13020291

**Published:** 2023-02-09

**Authors:** Eitan Elaad, Liza Zvi

**Affiliations:** 1Department of Psychology, Ariel University, Ariel 40700, Israel; 2Department of Criminology, Ariel University, Ariel 40700, Israel

**Keywords:** mood, Concealed Information Test, polygraph, self-assessed lying ability, orientation

## Abstract

The present study examined the effects of mood on physiological responses in the Concealed Information polygraph Test and the relation to self-assessed lying ability. One hundred and eight undergraduate students self-assessed their lie-telling ability, committed a mock theft, and were asked to conceal information related to the crime. Participants were then divided into three equal groups: two groups were asked to provide a detailed written account of either a happy or sad event in order to induce a happy or sad mood, while the third group served as a neutral group. Participants then underwent a polygraph test and were asked to try to avoid detection. An induced happy or sad mood tended to lower relative skin conductance responses to critical (crime related) items and enhance relative cardiovascular responses. Relative respiration responses to critical items obtained for the sad mood condition were more robust than the ones obtained for the happy mood condition. Under induced sad and happy moods, those who self-rated their lie-telling ability as high showed enhanced cardiovascular responsivity to critical items. These results were limited to the initial phase of the test. We discussed possible motivational explanations and implications for the Concealed Information polygraph test.

## 1. Introduction

The Concealed Information Test (CIT) is a psychophysiological detection method designed to detect information that an individual cannot or does not want to disclose [1]. The CIT presents examinees with a series of stimuli in which one critical stimulus is crime related, and the remaining stimuli function as controls. Enhanced physiological responses to the critical stimulus indicate the likelihood that examinees know about the crime. There is a general consensus among researchers that the standard CIT utilizes solid scientific principles and employs proper control questions. Laboratory research results reported high detection accuracy for both guilty and innocent examinees [1,2,3,4].

The CIT uses three physiological measures to detect potentially concealed information: electrodermal responses such as skin conductance response (SCR) amplitude, respiration changes such as the respiration line length (RLL) tracing, and cardiovascular responses such as heart rate (HR) or finger pulse waveform length (FPWL) [5,6]. A meta-analysis [7] that tested the efficiency of SCR amplitude on the CIT found it efficiently differentiates between informed guilty and uninformed innocent participants (Ben-Shakhar and Elaad reported a difference of 1.55 standard deviations). While CIT validity varies between studies, high CIT validity for skin conductance, respiration, heart rate, and p300 have been reported in a more recent meta-analysis [8].

Knowledge of the crime is known to be indicated by larger SCRs and attenuated respiration and cardiovascular activity, representing the examinee’s attention to and processing of the crime-related stimuli. The underlying mechanisms are the orienting response (OR) and response inhibition. The OR represents a group of behavioral and physiological responses evoked by any novel, unexpected, or personally significant stimulus [1,2]. As crime-related stimuli are personally significant to knowledgeable guilty suspects, they, therefore, evoke ORs, primarily in the form of increased SCR amplitude [3]. Guilty examinees try to suppress their physiological responses to crime-related stimuli, but such attempts produce the opposite effects of attenuated respiration and cardiovascular activity [4,9].

### 1.1. Impact of Mood

The CIT involves cognitive processes, such as memory detection, which require focused attention. As such, mood may affect the attention directed to critical (crime related) items by informed guilty suspects, and, therefore, may have practical implications in forensic assessment. For example, if either a happy or sad mood interferes with the attention directed to critical test items, polygraph examiners might attempt to moderate the examinee’s mood before the test; conversely, they may try to sustain the mood if it reliably helps predict the attention pattern of guilty examinees to critical test items. Thus, studying the question of whether mood affects CIT outcomes is essential. While no studies to date have directly examined whether the CIT’s measures are impacted by mood, we can glean some relevant information from previous studies on the impact of mood.

Early research [10] showed that happy and sad moods enhanced congruent memory. More recently, Forgas et al. [11] indicated that negative mood calls for accommodative processing, resulting in more focused attention on the precise details of the external world.

A positive mood, on the other hand, has been suggested to signal a predictive environment and calls for assimilative processing in which the individual relies on existing knowledge and heuristic thinking to perform a task [12]. Others have proposed that a good mood signals that a situation is favorable and little processing effort is required, whereas a bad mood recruits more systematic and vigilant processing [13]. Indeed, some have found that happy people may try and maintain their good mood by avoiding cognitive effort, while people in a bad mood may increase cognitive effort to improve their aversive mood [14]. Forgas and East [15] added that one might expect to find improved accuracy in eyewitness recollections when in a bad mood.

Forgas and East [16], also reported that mood biased the judgment of the genuineness of emotional expression. Finding that a sad mood increased skepticism, they indicated that mood may influence the ability to detect lies. Specifically, negative moods increased, and positive moods decreased, reported lie-detection ability. Although the present study’s focus is on being deceptive, rather than detecting deception, such findings underscore the impact that mood can have on the CIT.

In sum, the existing research cited above suggests that being in a negative mood is associated with increased attention to external stimuli and more effortful information processing. Therefore, mood might indeed affect examinee performance on the CIT, and interfere with memory detection. Specifically, we expected that guilty examinees in an unpleasant mood would focus their attention on the critical items of the CIT, such that their physiological responses may be amplified in ways that facilitate detection of concealed knowledge. In contrast, guilty—but happy—examinees would be less likely to pay such focused attention to the CIT’s critical items, and, therefore, have their knowledge be detected less often.

### 1.2. Self-Assessed Lying Ability

In order to isolate the impact of mood on CIT performance and detection, care must be taken to control for other related variables, such as an examinee’s self-proclaimed ability to lie convincingly. As mentioned above, CIT detection relies on the recognition of crime-related items by knowledgeable guilty suspects; while they may deliberately try to inhibit or suppress their physiological responses to critical items to avoid detection [9], these attempts may actually be related to their perception of their own lying ability. 

Self-assessed lie-telling abilities are often biased [17,18,19], where bias refers to a conscious or unconscious inclination that inhibits impartial judgment of the self. There are two kinds of biases. One bias associates the ability to deliver convincing lies with dishonesty (negative quality). People with this bias tend to score low on self-proclaimed lie-telling ability, to sustain a positive self-image. Thus, being a “poor liar” supports one’s view of oneself as basically honest [18]. Moreover, those who are captured by the difficult lie and simple truth bias find it difficult to influence others to believe their lies and have been found to rate their lie-telling abilities at the midlevel and sometimes even lower [17,18]. Another bias relates to the belief that the ability to lie successfully may serve one well. Relating to the possible advantages of being a good liar [20], people with this bias tend to highly rate their lie-telling ability. While we expected no higher-than-midpoint lie-telling-ability assessments, we expected to find a standard range of self-ratings from low to high among our cohort.

Elaad and Sommerfeld [21] found that performance on the CIT was indeed associated with self-assessed lying ability, reporting that lie-telling assessments predicted SCRs elicited by knowledgeable guilty participants to critical items. Specifically, guilty participants who provided higher ratings of their own ability to lie convincingly elicited more significant skin conductance responses to critical items than guilty participants reporting less confidence in their lying ability.

Elaad and Zvi [6] found that high scorers in self-assessed lie-telling ability used more frequent countermeasures on the CIT than low scorers. It follows that lie-telling ability assessments may contribute to the differential physiological responsivity. Specifically, high lie-telling ability scorers may be more anxious to prove their skills than low lie-telling ability scorers and, therefore, focus more on the crime-related items. Increased attention to critical items could possibly contribute to more significant physiological responses, mainly when examinees are in a bad mood.

### 1.3. The Current Study

In order to assess the impact of mood on performance on the CIT, and the association with self-assessed lying abilities, we utilized the Guilty Actions Test (GAT) questioning format [22,23]. The GAT format guarantees that “guilty” participants lie in response to critical items and tell the truth in response to control items. Accordingly, we designed a study to first have participants self-assess their lie-telling abilities and then instructed them on how to commit a mock theft. Subsequently, we divided the participants into three groups: two groups were asked to provide a detailed written report about a happy or sad event to induce a happy or sad mood, and the third group served as a neutral group.

Our hypotheses were as follows.

**Hypothesis 1.** *Mood manipulation would affect the examinee’s responses to CIT critical items. Specifically, a happy mood would interfere with CIT detection, while a sad mood would facilitate greater CIT detection*.

**Hypothesis 2.** 
*Self-assessed lie-telling ability high scorers would be focused on the critical items more than low lie-telling ability scorers. The enhanced attention would facilitate greater detection of their physiological responses, particularly in the sad mood condition.*


Nevertheless, as mood is a temporary and unstable state that may rapidly change, its strongest effect was expected to be exhibited during the initial stages of the CIT testing, before the mood-priming effect dissipated. Therefore, the CIT was divided into two blocks of questions to examine mood stability effects.

**Hypothesis 3.** 
*Mood effects would be more prevalent in the first CIT question block than in the second.*


## 2. Methods and Materials

### 2.1. Participants and Statistical Power

We recruited 108 undergraduate students (91 females) through internet ads for a polygraph experiment for course credit (mean age 22.8 years, SD = 1.7 years). The sample size (36 participants in each mood condition) related to a GPower analysis of effect size *f* = 0.4, with a power of 0.80 and α = 0.05. The study received approval from the research ethics committee of Ariel University and the participants provided their written informed consent.

### 2.2. Apparatus

The apparatus setup followed the procedure of Zvi & Elaad [24]. Participant responses were assessed utilizing three physiological measures: Skin conductance response amplitude (SCR), finger pulse waveform length (FPWL), and respiration line length (RLL). SCR amplitudes were recorded by a constant voltage system (0.5 V Atlas Researchers Ltd.) using two Ag/AgCl Grass electrodes of 0.8 cm diameter. FPWL responses were recorded using a piezoelectric plethysmograph (Atlas Researchers Ltd.) positioned around the right thumb. RLL responses were recorded using a piezoelectric device (Atlas Researchers Ltd.) hidden in the back support of the examinee’s chair and positioned at their thoracic level. A 19″ color monitor was positioned in front of the participants to present the questions and possible answers.

The air-conditioned laboratory used to conduct the experiment included an observation room separated from the laboratory via a one-way mirror through which monitoring could be performed. Two computers were used, connected by a serial communication link from a Data Acquisition System (DAS) that was split in parallel into the serial ports of the computers. One computer controlled the stimulus presentation and computed skin conductance, respiration, and cardiovascular changes. The second computer displayed a graphic form of the participant’s responses on a 19″ color monitor in the observation room; graph recordings were preserved for subsequent visual analysis and artifact control.

### 2.3. Procedure

A different experimenter conducted each of the two phases of the experiment, and they exchanged roles for approximately one half of the participants. The procedure is illustrated in the flowchart (Figure 1).

#### 2.3.1. First Phase of the Experiment

The participants met the first experimenter, who explained that the research involved undergoing a polygraph test to assess their abilities to avoid detection of critical information they possessed. The experimenter also informed the participants that they were eligible for two course credit hours, one for participation in the experiment and one for avoiding detection by the polygraph.

After obtaining consent to participate, the experimenter collected the participants’ personal information (e.g., gender and age) and asked them to self-assess their lie-telling abilities using the following question: “Compared to other people, how would you rate your ability to lie without getting caught?” They were asked to rate their ability on a 7-point scale ranging from 1 (much worse than others) to 7 (much better than others), with a midpoint of 4 (as good as others).

Subsequently, participants were asked to simulate a “guilty” condition by performing a mock theft. The participants were asked to select one of six folded instruction sheets and to follow its instructions to commit a mock theft in a designated office near the psychophysiological laboratory (where the polygraph test would take place). Each participant randomly received one out of four alternative crime profiles, adopted from an earlier polygraph study [25]: (a) Removing a book from a shelf in order to steal a *yellow* envelope addressed to the *research authority* containing *NIS 57*, a *necklace*, a *calculator*, and a *photograph of a bear*; (b) stealing a *green* envelope addressed to the *computer department* containing *NIS 24*, a *ring*, a *pen*, and a *photograph of a rhinoceros,* after removing a briefcase; (c) stealing a *red* envelope addressed to the *security officer* containing *NIS 49*, *earrings*, *spectacles*, and a *photograph of a zebra*, after removing a coat; and (d) stealing a *blue* envelope addressed to the *academic secretary* containing *NIS 63*, a *bracelet*, a *key*, and a *photograph of a lion,* after pushing aside a telephone (the critical items are marked in *italics*). The instructions informed participants to conceal the envelope’s content—the jewelry item, the sum of money, the object, and the photograph of an animal—in their pocket or bag and, then, proceed to the office where the experimenter was waiting.

Upon their return, participants were randomly assigned to either a sad, happy, or neutral mood-inducing condition. Participants assigned to the happy and sad mood conditions were asked to provide a vivid written report, with as many details as possible, about a happy or sad life event (depending on their condition) [11,26]. Participants in the neutral mood group were asked to write a report describing an evening of watching TV at home. After completing their report, the participants were asked about their current mood, using the following two questions: “At this very moment, I feel…?” (ranging from 1—sad to 9—happy), and “Now I feel…?” (ranging from 1—bad to 9—good).

#### 2.3.2. The Second Phase of the Experiment

After completing the first phase of the experiment, the participants entered the polygraph examination room, meeting a second experimenter who conducted the polygraph test. The second experimenter was aware of the participants’ “guilt” and the possibility of an induced mood (which were predetermined), but unaware of which mock theft profile had been randomly selected by the participants and, therefore, was also unaware of which set of critical items each participant was concealing. The second experimenter instructed the participants to sit in the examination chair, lean back into the back support, place their hands on the arm support, and refrain from moving during the entire test. While connecting the polygraph devices, the second experimenter told the participants they would undertake a polygraph test about stealing an envelope and reminded them that their task was to convince the experimenter that they had not been involved in the theft. The experimenter further reminded the participants of the opportunity to earn an additional hour of course credit if they passed the polygraph test. The experimenter then returned to the observation room. Further instructions were delivered through the intercom.

After an opening 2 min rest period, during which skin conductance baseline level was recorded, six CIT questions were communicated to the participants, each focusing on a separate item of the mock theft (the sum of money, the color of the envelope, the item of jewelry, the recipient to whom the envelope was addressed, the photograph, and the object). The questions and answer items were presented visually on a computer monitor and simultaneously narrated from pre-recorded sound files. The visual presentation (5 s for each item) began about 1 s before the onset of the auditory presentation (which lasted about 1 s). The inter-stimulus intervals ranged from 16 s to 24 s, with a mean interval of 20 s.

All the questions were presented in the GAT questioning format [22,23]. For example, participants were asked about the stolen sum of money: “Was the amount of money that you stole...?” After each question, 11 individual possible answers were presented (e.g., amounts of money), to which the participants were instructed to answer “no”. The first answer item was a neutral buffer item, in order to attract the initial orienting response. The buffer item was followed by two sets of the same 5 items, presented in a random order, with each set comprising the critical item (the correct answer, e.g., the exact sum of stolen money) and four unrelated control items (other sums of money).

After the first three questions of the test, a break was provided to allow the participants to rest and move their limbs. The break enabled the experimenter to remind participants, whenever necessary, about the instructions for the test period (e.g., avoid moving, do not cross your legs, do not yawn, sit up straight, etc.). After the break, the remaining three questions were presented. The questions were randomly ordered between participants.

At the end of the test, the participants were detached from the polygraph devices and left the examination room to return back to the office and the first experimenter. They were asked to complete, for the second time, the questions relating to their current feelings. The experimenter also asked the participants to recall the items they remembered from the mock theft and to indicate the extent to which they felt guilty or innocent during the experiment (manipulation check for guilt) on a scale from 0 *(utterly innocent*) to 100 *(utterly guilty)*. Participants were additionally asked to evaluate their level of excitement throughout the polygraph test on a scale ranging from 0 (*very little*) to 100 (*very much*) and to estimate how successful they were in avoiding the detection of concealed information during the test on a scale ranging from 0 (*not at all*) to 100 (*absolutely*), with a midpoint of 50 *(to some extent)*.

Finally, the experimenter debriefed the participants about the test and granted additional course credit to participants who had successfully avoided detection (participants were rewarded if they produced a larger SCR amplitude than the controls to critical items for no more than two of the six CIT questions). About one-third of the participants received the bonus.

## 3. Results

### 3.1. Manipulation Checks

#### 3.1.1. Mood

In the first phase of the experiment, after providing a written report of a happy, sad, or neutral event they had experienced, the participants described their current feeling by responding to questions that referred to their current mood: sad/happy and bad/good. The identical questions reappeared at the end of the polygraph test in the second phase of the experiment. Table 1 shows the mean ratings of the participant’s sad/happy and bad/good feelings (higher scores reflect stronger positive feelings and lower scores reflect stronger negative feelings) across the two experimental phases and *three* mood conditions. For technical reasons, current state data were available for only 22 participants in the sad mood condition, 29 in the good mood condition, and 33 in the neutral condition.

We conducted a 2 × 2 × 3 ANOVA on the current mood ratings with two within-subject factors Time (before and after the polygraph) and Questions (sad/happy, bad/good), and one between-subjects factor Mood (sad/happy/neutral). A significant Time × Mood interaction effect, F _(2,80)_ = 5.77, *p* = 0.005, η_p_^2^ = 0.13, suggests that the participant’s mood ratings were consistent with the mood condition to which they were assigned, but only before the administration of the polygraph test and not after the test (see Table 1). No other significant main or interaction effects were observed. This result suggests that the mood-inducing manipulation appeared to influence the participants in the happy mood condition to feel happy/good, and the participants in the sad mood condition to feel sad/bad. However, this result was not repeated after the experiment, suggesting we should treat mood as provisional.

#### 3.1.2. Guilt

Following the polygraph test, participants were asked to indicate the extent to which they felt innocent/guilty during the mock theft, on a scale ranging from 0 (*utterly innocent*) to 100 *(utterly guilty*). The purpose of this question was to ensure the participants understood they were “guilty” of stealing the hidden envelope. The findings showed that the guilt manipulation was effective, and the participants realized they were guilty (mean = 71.6, SD = 32.3). A one-way ANOVA on the guilt ratings for the three mood-inducing conditions showed no significant difference, F _(2,104)_ = 0.003, *p* = 0.99 (for one participant, the information was not available).

#### 3.1.3. Excitement

The participants expressed moderate excitement during the test (mean excitement = 59.3, SD = 23.6), which was very similar to the excitement expressed in previous laboratory CIT studies [5]. A one-way ANOVA performed on the excitement ratings for the three mood-inducing conditions revealed no significant mood differences, F _(2,105)_ = 1.73, *p* = 0.18. The mood conditions did not affect excitement.

#### 3.1.4. Success Assessments

The participants assessed their success in avoiding the detection of their concealed information on a scale ranging from 0 (*not at all*) to 100 (*absolutely*). The results showed that the mid-point 50 (*to some extent*) best represented the participants’ assessments (mean = 50.5, SD = 21.5). We performed a one-way ANOVA to detect possible mood differences in avoiding the detection of concealed information. The results indicated that the three mood-inducing conditions did not significantly affect their assessed success avoiding detection, F _(2,105)_ = 0.119, *p* = 0.89.

#### 3.1.5. A Memory of the Critical Items

After the experimental session, the participants completed a recall test in which they were asked to write down all the six critical items that had appeared on their mock crime instruction sheet. The mean number of critical items they remembered was 5.12 (SD = 1.15). This high recall rate is consistent with the number of recalled items found in similar laboratory studies [5,25].

### 3.2. Physiological Response Scoring and Analysis

We converted all the physiological responses into within-subject standard scores relative to the respective means and standard deviations related to each multiple-choice question containing eleven items. The critical stimuli Z-scores incorporate response comparisons to control items. Furthermore, within-subjects Z-scores eliminate individual variations in responsivity and allow a meaningful summation of the responses of different participants. Standard scores were computed for each physiological measure (SCR, FPWL, and RLL) separately, as follows:

#### 3.2.1. Skin Conductance Responses

SCRs were transmitted in real-time to the computer. SCR was defined as the maximal increase in skin conductance from 1 to 5 s after stimulus onset. The sampling rate for SCRs was 20 Hz.

#### 3.2.2. Finger Pulse Waveform Length

The following description of the FPWL responses was adopted from a previous study that first used the responses [27]. “FPWL responses were defined as the total pulse pattern line length that depicts the activity of the peripheral blood vessel during a 15-s interval following stimulus onset. The line length of the pulse pattern is disproportionally affected by the starting point of measurement. Therefore, ten 15 s windows were created, each beginning 0.1 s later than the previous one, and the FPWL was defined as the mean of these ten length measures. The sampling rate for FPWL was 20 Hz. The FPWL combines pulse rate slowing and a decrease in pulse amplitude. Therefore, the FPWL response is reflected by a shorter line length—the shorter the line, the stronger the response. Z scores were computed relative to the mean and standard deviation of the examinee’s FPWL responses within every question of eleven items and multiplied by −1.”

#### 3.2.3. Respiration Line Length

RLL responses were measured by the device hidden in the chair back support and were defined by the total respiration line length during a 15-s interval following stimulus onset, in which shorter lines correspond to more robust responses. Following Elaad et al. [28]: “Each response was defined as the mean of 10 length measures (0.1 s through 15.1 s after stimulus onset, 0.2 s through 15.2 s after stimulus onset, etc.). This way, ten 15 s windows were created, each beginning 0.1 s later than the previous one, and the RLL was defined as the mean of these ten length measures. Each RLL was computed using a sampling rate of 20 Hz. All RLL Z scores were multiplied by −1.”

### 3.3. Association between the Physiological Responses

SCRs are primarily the outcome of short-term orientation responses (OR), whereas FPWL and RLL responses result from prolonged attention or deliberate response inhibition [9,24]. Therefore, RLL and FPWL would be strongly associated. Pearson correlations computed for the three physiological measures (N = 108) indicated that the correlation between FPWL and RLL was the largest, r = 0.392, *p* < 0.001. In comparison, the correlations between SCR and FPWL, r = 0.245, *p* = 0.011, and SCR and RLL, r = 0.191, *p* = 0.048, were still significantly positive but lower.

### 3.4. Comparing the Detection Efficiency of the Physiological Responses

We computed the mean standardized responses to the critical items for each participant across the six CIT questions and each block of three questions: Block 1 (the first three questions before the break) and Block 2 (the final three questions). Mean scores across participants served as the CIT detection score. Physiological data were gathered and analyzed irrespective of the participant’s recall of crime-related items to avoid increased detection efficiency. Table 2 presents each block’s mean Z scores for each physiological measure.

We performed a 3 × 2 ANOVA with repeated measures to assess whether the three physiological measures (SCR, FPWL, RLL) and the two question blocks (first, second) differed in their effects on the critical items’ Z scores. After correcting for sphericity (ε = 0.866), a significant measure effect emerged, F _(1.7,185.3)_ = 7.0, *p* = 0.001, η^2^_p_ = 0.06, indicating that RLL showed the lowest differential response magnitude to the critical items (see Table 2). However, in line with previous studies [24], the question blocks showed no significant effect on the relative magnitude of the responses to the critical items.

### 3.5. Hypotheses Testing

We hypothesized that participants in a sad mood would respond more strongly to critical items than participants in either a happy or a neutral mood, at least in the initial phase of the test (Block 1) before the mood effect might potentially fade.

#### 3.5.1. Mood and Physiological Detection in the CIT

A 3 × 3 ANOVA with repeated measures was performed on the mean Z-scores for Block 1, with Mood (the three mood-inducing conditions) as the between-subject factor and Measures (SCR, FPWL, and RLL) as the within-subject factor. The analysis showed no significant main effects (Mood: F _(2,105)_ = 0.45, *p* > 0.05, η^2^_p_ = 0.009, and Measures: F _(2,105)_ = 0.24, *p* > 0.05, η^2^_p_ = 0.002). However, a significant interaction effect emerged, F _(2,105)_ = 4.63, *p* = 0.012, η^2^_p_ = 0.08. Table 3 shows different mood effects on the three physiological measures.

We separated the three physiological measures and found that only the results for respiration responses were in the expected direction (sad mood increased, and happy mood decreased, respiration responses). However, a one-way ANOVA conducted to demonstrate the mood effect on RLL responses failed to show it, F _(2,105)_ = 1.50, *p* > 0.05, η^2^_p_ = 0.028. A *t*-test that compared the two mood-inducing conditions (sad, happy) according to the one-sided hypothesis that a sad mood would elicit more significant respiration responses than a happy mood confirmed it, meaning t _(36)_ = 1.76, *p* = 0.042, d = 0.41.

The results for SCR and FPWL responses failed to support the hypotheses. Still, research on potential mood effects on the CIT is in its infancy, and the present study is the first to accept the challenge. The lack of other results leads us to consider possible mood effects that were not hypothesized from the beginning, but are suggested by the data. The SCR results showed that the two mood-inducing conditions (sad/happy) had lower detection rates than the control condition. As SCRs are highly sensitive to orientation effects, these results suggest that mood may interfere with OR release and impair SCRs to critical items. According to the one-sided interference assumption, SCRs elicited in the sad and happy mood-inducing conditions were combined and compared to SCRs elicited in the control condition. A one-tailed *t*-test revealed a significant difference, t _(106)_ = −1.91, *p* = 0.03, d = 0.39, indicating possible mood interference with detecting critical knowledge by SCRs. Still, the effect is small and needs further support.

FPWL responses showed an inverse effect (Table 3). Specifically, critical items in the mood-inducing conditions were better detected than critical items in the neutral condition. This difference may suggest that mood facilitates FPWL detection in the CIT. FPWL responses are less sensitive to orientation and more dependent on prolonged attention than SCRs. Therefore, deliberate attention to critical CIT items may guide cardiovascular responses, and they are more resistant to interference with the release of ORs. Subsequently, we combined two of the mood-inducing conditions to examine the post hoc assumption that mood manipulations, either sad or happy, contribute to detection by FPWL responses. We then compared them with the neutral mood condition. A significant difference was found, t _(106)_ = 1.83, *p* = 0.035, d = 0.37. Nevertheless, the effect size is small. We observed no mood effects on physiological responses in the second block of CIT questions.

#### 3.5.2. Signal Detection Model—ROC

To further examine the detection efficiency of the three physiological measures, we used another method derived from signal detection theory, namely, the receiver operating characteristic (ROC) model. Earlier research employed the ROC model in CIT studies [29,30,31,32,33]. Further, the National Research Council Report [34] endorsed the ROC model as an appropriate method for depicting the diagnostic value of polygraph tests.

The ROC model is relevant to detection efficacy. It defines the amount of separation between the distributions of the responses to the critical items produced by participants assigned to experimental and control conditions. The present study did not use an actual control group (uninformed innocent participants), but, rather, used a simulated innocent condition [35]. The advantage of a simulated control condition over an actual control group is the ability to use the same standard for all comparisons between experimental and simulated conditions. The simulated uninformed group was formed by randomly drawing eleven values from a standard normal distribution (mean = 0, standard deviation = 1). The drawing was repeated six times for each participant (simulating the six CIT questions) and was then averaged to represent the score of one uninformed control participant. In this way, we created a control group of 108 simulated innocent participants.

We calculated the mean Z score distributions for the experimental groups (sad/happy/neutral) across the six items for each physiological measure. ROC (Receiver Operating Characteristic) curves were then generated based on the distributions of the experimental and the simulated neutral groups.

Following Elaad and Zvi [6], “The areas under these ROC curves, and the corresponding 95% confidence intervals, were computed [36]. The area statistic describes the participant’s detection efficacy across all possible cutoff points. The assumed values range between 0 and 1 so that an area of 1 indicates a complete separation between the two distributions. In contrast, an area of 0.5 indicates a perfect overlap of the two distributions.” Table 4 displays the area under the ROC curves and the corresponding confidence intervals computed for each physiological measure (SCR, FPWL, RLL). Table 4 shows that all ROC areas computed for the various indices were significantly larger than chance (the lower bounds of the 95% confidence intervals computed for ROC areas are no less than 0.5). Specifically, the experimental groups had significantly higher responses to the critical items than the corresponding simulated control groups. These significant results were relevant to all individual physiological measures.

#### 3.5.3. Self-Assessed Lie-Telling Abilities

Statistics for the self-assessed lie-telling abilities were computed for the entire sample (N = 108) as follows: Mean = 3.87, SD = 1.52, 95% Confidence Intervals = 3.57–4.16 (Confidence Intervals are based on standard error units). The results suggest that the mean is not significantly different in self-assessed lie-telling abilities from the scale midpoint score four. Outcomes are consistent with previous results (see [18] for a review).

#### 3.5.4. High Lie-Telling Ability Scorers and Mood Effects in the CIT

As mood and self-assessed lie-telling ability are related, and the combined effect of high self-assessed lying ability and sad mood may be associated with physiological responsivity, physiological responses of those who self-rated their lie-telling ability as high (higher than 4 on the 7-point scale) were separately analyzed. Table 5 presents these responses within the three mood-inducing conditions. Only the first block of three CIT questions, in which evidence of the induced mood remained, was examined. We performed a one-way ANOVA with one between-subjects factor—Mood (the three mood-inducing conditions: happy, sad, neutral) for each physiological measure. A significant mood effect emerged for FPWL, F _(2,39)_ = 3.47, *p* = 0.041, η^2^_p_=.15. Table 5 shows the enhanced FPWL responsivity under the happy and sad induced mood conditions, which may indicate that the mood manipulation (sad and happy) contributed to enhancing FPWL responses. We then compared the combined happy and sad induced mood conditions to the neutral condition, and the results indicated that FPWL responses in the combined mood condition were significantly larger than those in the control condition, t _(40)_ = 2.34, *p* = 0.012, d = 0.75.

Another perspective is offered by Bayesian statistics, which indicates the ratio of the likelihood that we would obtain the observed results under a true null hypothesis compared to a true alternative hypothesis. Bayesian independent-sample *t*-testing conducted on FPWL responses showed that BP_01_ = 0.43. Thus, the test hypothesis is six times more probable than the null hypothesis (no difference exists between mood conditions). This suggests that mood facilitates cardiovascular detection on the CIT among those who scored themselves high on lie-telling ability in the context of sad and happy induced moods. The results obtained for RLL (Table 5) were not significant (F _(2,39)_ = 0.63, *p* > 0.05). Similar non-significant differences were obtained for SCRs (F _(2,39)_ = 0.63, *p* > 0.05).

## 4. Discussion

Our first hypothesis, which stated that inducing happy, neutral, and sad moods would differentially affect examinees’ responses to CIT critical items, was not conclusively validated or rejected. This may be partly due to the mood manipulations utilized in the study being insufficiently effective, and of limited duration, but was more likely due to the interaction between mood effects and the various physiological responses. Nevertheless, the interaction provides new insight into the influence that mood may have on CIT performance.

The results suggest that mood may lower SCRs and enhance FPWL responses to critical items. In contrast, RLL responses showed the expected direction, and we observed more robust responses to critical items in the context of a sad mood than in a happy mood. Differences in orientation and prolonged attention or arousal inhibition in the CIT may explain the outcomes.

ORs help explain differential physiological responses to critical and control CIT items because the critical item has special significance to a knowledgeable person and generates more significant ORs than the control items [3]. It is widely accepted that SCR amplitude is the best autonomic indicator of OR [37]. On the other hand, RLL slowing and heart rate (HR) deceleration are less sensitive to OR influences [4]. Two studies [37,38] have demonstrated that SCR is mainly a measure of orienting, whereas RLL and HR responses are measures of arousal inhibition [9]. Arousal inhibition theory suggests that deliberate attempts to inhibit physiological arousal underlie the CIT. Specifically, a desire to conceal information in the CIT encourages deliberate attempts to inhibit physiological responses, which paradoxically contributes to more significant, rather than smaller, physiological responses to the critical items.

In light of this, we propose that mood, particularly a sad mood, interferes with the release of ORs and, at the same time, encourages prolonged attention and deliberate inhibition of physiological responses. Additional research is necessary to lend support to this notion.

Another factor that may be related to CIT detection is the self-assessed lie-telling ability. The mean lie-telling ability rating (3.87) was not significantly different from the midpoint score of 4. Such an outcome is consistent with previous results [18]. Previous studies indicated gender differences in lie-telling ability assessments, e.g., women rate their lie-telling ability lower than men. Gender differences have been found for students [39] and community members [40]. Nevertheless, men in our study did not overestimate their lie-telling ability.

Of particular interest were participants who rated their lie-telling ability as high, likely perceiving the advantages of being a successful liar [20]. Recent research has indicated that self-assessed good liars also report a greater number of lies told on a daily basis, communicating more lies in face-to-face interactions, and providing plausible accounts for their lies [41]. Furthermore, self-reported good liars tend to use more countermeasures in the CIT [6] and may feel competent to influence the polygraph outcome [21].

The association between higher lie-telling ability self-assessments and induced mood resulted in more significant physiological responses and better detection of critical CIT items, particularly from FPWL responses, which are more sensitive to prolonged attention than SCRs. Such results might be explained by greater motivation among self-assessed high scorers in lie-telling ability to prove their superior lying skills in the context of challenging, daunting polygraph tests. Therefore, they may have focused greater attention on the critical items while lying, which generated more robust cardiovascular responses to the critical items and facilitated greater detection. However, this interpretation should be seen as preliminary until further experimental evidence that supports it has been collected.

Notwithstanding, the association found between mood and self-perceived lying ability has potential significance from an applied perspective, if further confirmed. Polygraph testing in real life situations is often characterized as a highly fraught, stressful experience in which suspected individuals (guilty and innocent) must, in effect, “prove” their innocence or face potentially severe consequences. Accordingly, it is probable that such an unpleasant testing experience may induce a negative mood state. Indeed, the intensity of a bad mood evoked in the context of an actual polygraph is presumably much higher than that generated by the present manipulation. Given that the present results suggest that higher confidence in one’s lying ability facilitates greater CIT detection when an examinee is experiencing a bad mood, a more intense negative mood induced in real life polygraph testing may lead to even greater detection depending on one’s self-assessed lying ability. Notwithstanding, the present results are not yet relevant to actual polygraph testing, as future research is needed to confirm or refute them under more effective and lasting mood conditions.

### Limitations and Future Research

Several limitations should be noted. The mood manipulation may have been ineffective and limited in duration, as the results appeared to show traces of mood manipulation effects only in the first block of questions; therefore, our analysis was limited to the initial phase of the test.

Further, the participants reported experiencing a fairly moderate level of excitement during the CIT. Accordingly, the experimental situation does not appear to resemble the arousal expected of examinees in real-life situations in which the stakes are much higher and excitement is expected to be more intense. Moreover, since participants regarded the situation as an experiment with relatively minor consequences (i.e., an hour of course credit), inducing moods was rather tricky. Due to these limitations, the results should be interpreted with caution. Further research is necessary to corroborate the present findings and would benefit from including other factors potentially affecting the CIT, such as stress levels, stress appraisal, personal values, motivation to succeed, attention direction, and other important individual traits.

The present sample of participants consisted mainly of young female students. Although females are no more or less inclined to respond to critical items on the CIT [21], they have tended to rate their lie-telling ability somewhat lower than males, although this difference may not necessarily have reached significance or been the product of more than a small effect size [18]. Furthermore, studies comprising a male majority [20] have displayed similar lie-telling ability assessments as those using a female majority [21]. Thus, the female majority in the present study does not inherently bias the results. Still, as most real-life CIT examinees are male suspects from the general population with a diverse age and education distribution, the current cohort’s different composition dictates that caution be used when considering the external validity of the present results. Future studies are advised to use a more equal ratio of men and women and include a more comprehensive age range in order to further examine the potential effects and interactions of mood, gender, and age on CIT performance. Finally, the simulated control participants may raise questions about their similarities with actual innocent participants.

Replication is necessary to clarify the present results. In the replication, it is advised to enhance the strength and longevity of the mood manipulation by talking about a happy or sad life event which may be more effective than writing about it. Furthermore, the sample size of the current study (36 participants per condition) was too small to detect the weak mood effects. Therefore, future studies may benefit from a larger sample size. In addition, future research should use actual uninformed innocent participants to control for the experimental mood conditions. Finally, future studies are advised to keep a balanced sample of men and women from a more comprehensive age and education range to resemble the population examined by the polygraph.

## 5. Conclusions

The present study examined how transitory emotional states (mood) affect physiological detection on the CIT and how the related self-assessed lie-telling ability explains physiological responses. The guilty examinees’ level of self-confidence in their ability to tell lies convincingly may be associated with more significant FPWL responses in the context of a sad mood. However, more research is needed to support this notion.

The present results should inspire additional research on the effects of mood in the context of performance on the Concealed Information Test. Additional research is essential because of the severe shortage of similar empirical studies and the possibility of significant implications for field practice. As people vary tremendously in their cognitive and affective styles, dispositional capacity for emotion, reactivity to the environment, and ability to regulate emotions [42], these and other individual differences may influence mood and examinees’ performance on the CIT. Therefore, they deserve research attention.

## Figures and Tables

**Figure 1 brainsci-13-00291-f001:**
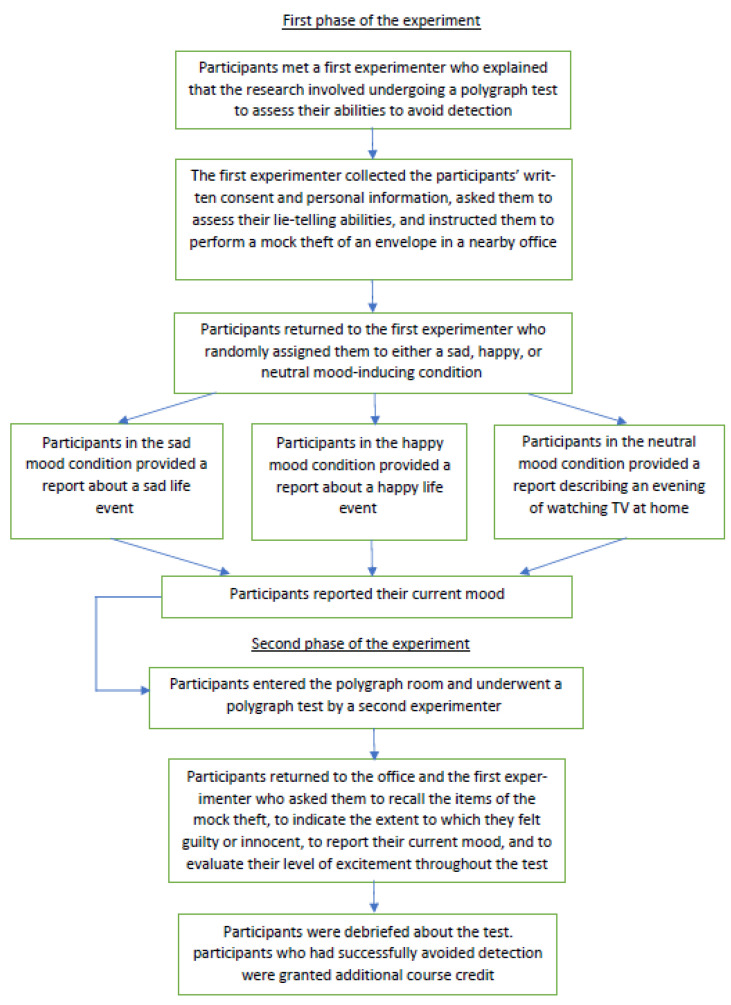
Flowchart of the procedure.

**Table 1 brainsci-13-00291-t001:** Means (and SDs) of Participant’s Happy/Sad and Good/Bad Feelings in Three Mood Inducing Conditions, Before and After the Polygraph Test.

Feelings	Sad Mood	Neutral Mood	Happy Mood
Happy/Sad before	4.50 (2.04)	6.00 (1.48)	6.28 (1.71)
Happy/Sad after	6.00 (1.00)	6.12 (1.29)	6.03 (1.76)
Good/Bad before	6.00 (1.60)	6.21 (1.97)	7.17 (1.73)
Good/Bad after	7.14 (1.59)	6.85 (1.52)	7.17 (1.20)
N	22	33	29

*Note*. Higher scores indicate a more positive mood.

**Table 2 brainsci-13-00291-t002:** Means (and SDs) of Z Scores Computed for Each Physiological Measure Within and Across the two Question Blocks.

Block	SCR	FPWL	RLL
Block 1	0.44 (0.54)	0.41 (0.42)	0.28 (0.42)
Block 2	0.41 (0.61)	0.42 (0.41)	0.25 (0.46)
Across	0.42 (0.49)	0.41 (0.34)	0.27 (0.36)

*Note*. High scores indicate significant responses. SCR—Skin conductance response amplitude. FPWL—Finger pulse waveform length. RLL—Respiration line length.

**Table 3 brainsci-13-00291-t003:** Means (and SDs) of Standardized SCR, FPWL, and RLL Responses to Critical Items in the First Block of Questions Under the Three Mood-Inducing Conditions.

Measure	Sad Mood	Neutral Mood	Happy Mood	Across
SCR	0.39 (0.53)	0.58 (0.54)	0.34 (0.54)	0.44 (0.54)
FPWL	0.46 (0.43)	0.31 (0.43)	0.47 (0.39)	0.41 (0.42)
RLL	0.37 (0.45)	0.28 (0.43)	0.20 (0.36)	0.28 (0.42)
N	36	36	36	108

*Note*. Higher scores indicate larger responses. SCR—Skin conductance response amplitude. FPWL—Finger pulse waveform length. RLL—Respiration line length.

**Table 4 brainsci-13-00291-t004:** Areas under the ROC curves and 95% confidence intervals computed for the three physiological measures.

Measure	Area	SE	95% CI
SCR	0.75 *	0.039	0.67–0.82
FPWL	0.82 *	0.033	0.76–0.89
RLL	0.75 *	0.038	0.68–0.83

* *p* < 0.001. SCR—Skin conductance response amplitude. FPWL—Finger pulse waveform length. RLL—Respiration line length.

**Table 5 brainsci-13-00291-t005:** Means (and SDs) of Z scores among high lie-telling ability scorers computed for each physiological measure and mood condition in CIT question block 1.

Measure	Sad Mood	Neutral Mood	Happy Mood
SCR	0.52 (0.49)	0.47 (0.41)	0.35 (0.37)
FPWL	0.64 (0.33)	0.22 (0.48)	0.45 (0.39)
RLL	0.37 (0.40)	0.19 (0.51)	0.31 (0.39)
N	12	15	15

*Note*. Higher scores indicate larger responses. SCR–Skin conductance response amplitude. FPWL—Finger pulse waveform length. RLL—Respiration line length.

## Data Availability

Additional data will be provided on request.

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
