# Peer review of "Effects of Mood on Psychophysiological Detection of Concealed Information and the Relation to Self-Assessed Lying Ability"

_brainsci, 2023, doi:10.3390/brainsci13020291_

Round 1
Reviewer 1 Report
Review of Manuscript #brainsci-2155206 submitted to Brain Sciences:
Effects of mood and self-assessed lying abilities on psychophysiological detection of concealed information
January 11th, 2023
The purpose of this submission was to study the effect of inducing happy, sad, or neutral moods on participants’ physiological responses while telling a lie during the Concealed Information polygraph Test (CIT). The authors predicted that induced sad mood would augment the detection of lies using the CIT, by increasing the physiological responses to test questions about deception. Results provided mixed support for the hypotheses. Contrary to the prediction, both induced happy mood and sad mood lowered skin conductance responses and enhanced cardiovascular responses to critical test questions, as compared to the neutral mood condition. As predicted, respiration responses to critical test questions were greater under the sad mood condition than the happy mood condition. People who rated themselves high in lie-telling ability showed enhanced cardiovascular responses to critical test questions when in a bad mood, as compared to the other two mood conditions.
This appears to be the first study to examine how the CIT polygraph is affected by participants’ mood. Thus, the results extend knowledge regarding the efficacy of the CIT in detecting deception. While the hypotheses received only partial support, this is a promising new direction for polygraph research.
Most importantly, I believe this manuscript could benefit from clarifying the reasons for some of the more puzzling results, as well as focusing on exactly how future research can help clarify the findings.
To the authors’ credit, they clearly did not attempt to pretend that the findings were exactly what was predicted; there was a clear and forthright effort to state which results were predicted and which results were unexpected. Nevertheless, some of the results are puzzling. Why should sad mood enhance enhance responses to critical items for some physiological cues, while both happy and sad mood enhanced responses to critical items for other physiological cues? The authors mention some plausible explanations briefly in the discussion, but this is speculative. Ideally, the authors would conduct a full replication, ideally with a larger sample size, to confirm the findings. I realize this is probably not practical, so the next best thing would be to provide a detailed plan for identifying exactly what is needed to clarify the present results in future studies. Should future research focus on enhancing the strength or longevity of the mood manipulation? (Perhaps talking about a happy or sad life event would be more effective than writing about it?) Or does it make sense to focus only on cardiovascular cues, or only respiratory cues? And will the results only be found in self-proclaimed good liars? In other words, the discussion section could benefit from a detailed blueprint for the next logical study, as well as specific predictions to help clarify the current findings.
The sample size for the current study (N = 108, or N = 36 participants per condition) is a little smaller than would be ideal. When discussing potential flaws in the discussion section, it would be useful to mention that future studies may benefit from a larger sample size.
The abstract stated that self-proclaimed good liars showed enhanced cardiovascular responsivity to critical items when in a sad mood. However, if I understand the results section correctly, the results on p. 12 indicate that self-proclaimed good liars showed enhanced cardiovascular responsivity to critical items when in a sad or good moods, as compared to neutral mood. Is it that bad mood showed the most responsivity, followed by good mood, and finally neutral mood? This result should be clarified, and the description of the result should be consistent between the abstract and section 3.5.4 on p. 12.
I have just a few other minor issues to mention:
In the middle of p. 5, the authors mentioned that the second experimenter was not blind to the participants’ induced mood. Given that this second experimenter was in charge of measuring key dependent measures, is there a reason why she/he was not blind to conditions?
The authors mention that mood states did not affect excitement (p. 7). However, the F-value was not close to zero. The p-value is labeled as “> .05.” However, the lack of a significant result does not conclusively indicate that a null result is definitely correct; rather, a non-significant p-value only indicates that one cannot conclude an effect does exist. It would be helpful to include exact p-values on this page, rather than “p > .05”.
In the tables on p. 10 and p. 12, it would be helpful to include subscripts next to the means in each row to indicate which means differ significantly from each other. For example, means with no significant differences would have the subscript “a,” while a significantly different mean within a given row would be labeled with the sunscript “b,” etc.
I’m not convinced that the simulated control participants (described on p. 11) would necessarily behave similarly to actual participants, so I think it is good that authors don’t discuss this result too extensively.
At the bottom of p. 11, the authors state that the self-assessed lie-telling ability of 3.87 out of 7 is not significantly different than the “average score,” or 4 out of 7. Rather than labeling 4 as the “average score,” I think the term “scale midpoint” would be more accurate. (The midpoint of a scale is not necessarily where the “average” person would answer.)
Some of the mood states have multiple labels – e.g., “sad” is sometimes called “bad,” and the terms “control” vs. “neutral” appear to be interchangeable. I think it would be helpful to use a single term throughout the paper for each mood state.
In sum, this manuscript adds substantively to the literature on the efficacy of the CIT polygraph. Although not all of the hypotheses were supported, it is important for researchers to understand how target people’s moods may affect their polygraph scores. The manuscript could benefit from clarifying the reasons for some of the more puzzling results, as well as focusing on exactly how future research can help clarify the findings with a large-N preregistered study.
Reviewer 2 Report
The study examined the effects of mood (good, bad, control) and self-determined lying ability on various physiological measures in a concealed information polygraph test (CIT). The authors induced a good or bad (or control) mood by asking the participants (Ps) to write a detailed report of a pleasant or unpleasant experience. The Ps also rated themselves on their ability to lie effectively. The goals of the study were to explore the possible interaction between mood and (self-defined) lie-telling ability. Depending on whether one takes the authors’ claim, or the results of the formal statistical analyses, the patterns of results either showed that mood and self-assessed lying ability interacted with each other (authors’ claim) or did not interact with one another (results of formal statistical analyses).
I found it difficult to understand the methodology, in part, because I am not well versed in CIT research and, in part, because the writing was unclear. For instance, the paper states (p. 5) “ Each question presented 11 items: one critical item … and four unrelated control items….” So, I wasn’t clear whether each question presented 11 items, or each question presented 5 items (1 + 4). [To add to the confusion, at times I was unsure about the differences among the concepts of “items” and “questions,” as the terms seemed to be used interchangeably.]
Similarly, the authors give an example of each question (p.7): “Was the amount of money that you stole 75 NIS, 63 NIS…? The participants answered “no” to each question.” I take that to mean that P answered “no” to the question “Was the amount of money you stole 75 NIS?” and also answered “no” to the parallel question “Was the amount … 63 NIS?” I.e., P answered “no” to every question – which, in itself would seem odd, if P had to give the same answer to every question. But later, (p. 9, line 330), the author describes each question as “multiple-choice,” –which seems to suggest that all 5 options were provided, and P was supposed to select one of the presented choices. But then, what does it mean for P to say “no” if P is supposed to select from one of the provided options?
I think it would have helped the reader—I’m sure it would have helped me—if the authors gave a concrete example of each concept, including items, questions, irrelevant items, unrelated items, control items. Perhaps the terminology would be clearer for readers who are more familiar with CIT testing, in which case the writing may not be problematic – at least for readers who are knowledgeable about CIT testing. But I would hope that some readers would have minimal background in CIT research.
On the more conceptual side, the authors’ first hypothesis (bad mood will yield stronger signs of deception) was not supported by the data (no main effect of Mood). The authors do find an interaction between mood and physiological measure, but that interaction was not predicted, and so any attempts to explain the interaction are post-hoc.
The first major hypothesis, that being in a bad mood—in comparison to being in a good mood—would strong deception cues. I found it odd, therefore, that a later analysis was conducted that combined good and bad mood and compared it to being in a neutral mood. I did not see how that comparison was relevant to the story that was being developed in the Introduction.
The other major hypothesis (# 2) was that self-assessed lying ability would interact with mood: High self-assessed liars would focus on critical items more than low self-assessed liars, particularly in the bad mood condition. Obviously, such an interaction mandates at least a 2 x 2 design, with high- and low-self self-assessed liars and being in a bad or good Mood. But the data (section 3.5.4) are examined only for high self-assessed liars. That obviously will not allow for a test of an interaction. It was not clear to me why, if the goal was to examine an interaction, the authors examined and presented the data from only high self-assessed liars.
Finally, the authors analyze the data to see if there is a meaningful effect of Mood on each of the three physiological measures (FPWL, RLL, and SCR). It was not clear to me why the authors separated the data for the three physiological measures. I had a few concerns about this analysis. First, the authors combine the scores for good and bad mood and compare those combined mood scores with the neutral mood. Again, no compelling justification is provided for combining good and bad mood scores, and doing so seems to violate one of the basic goals of the study, which is to distinguish between good and bad mood. (How does merging the two scores help to distinguish between them?). Second, the effect of Mood is significant for one measure (FPWL) but not for the other two measures (RLL and SCR), although the authors note that the scores were “in the same direction” for the RLL and SCR measures as for the FPWL measure. I note that the F scores for the RLL and SCR measures were both less than 1, which is quite low to be claiming that the scores “were in the same direction.”
A few minor comments.
1. The term GAT is not defined the first time it is used.
2. The authors use the language of causality when discussing the role of self-assessed lying, but as the data are observation (not controlled experimentally), it may be better to speak about self-assessed lying as being correlated with (rather than causing) some other behavior.
3. The authors seem to give some weight to the finding that the mean ” score on the self-assessed lying is below the “average”—I assume the middle of the scale. As the scale is not measured in countable units, but is somewhat arbitrary, I’m not sure that much can be made of the finding that the mean score was “below average.”
